# Enhancing COVID-19 Forecasting Precision through the Integration of Compartmental Models, Machine Learning and Variants

Daniele Baccega*
daniele.baccega@unito.it
University of Turin
Turin, Italy

Paolo Castagno
paolo.castagno@unito.it
University of Turin
Turin, Italy

Antonio Fernández Anta
antonio.fernandez@imdea.org
IMDEA Networks Institute
Madrid, Spain

Matteo Sereno
matteo.sereno@unito.it
University of Turin
Turin, Italy

## Abstract

Predicting epidemic evolution is essential for making informed decisions and implementing effective countermeasures. Computational models provide valuable insights into disease progression, enabling early detection, timely intervention, and effective prevention strategies. These models help allocate resources and protect public health by anticipating the course of an outbreak and allowing for proactive measures. We propose Sybil, a framework that merges machine learning with variant-aware compartmental models, combining data-driven and analytical methods. We tested Sybil's predictive capabilities using COVID-19 data from Italy, Austria, and U.S., including records of new and recovered cases, fatalities, and the presence of different variants over time. Our evaluation focused on Sybil's forecasting accuracy during periods of significant trend changes. The results indicate that Sybil surpasses traditional data-driven approaches, accurately predicting trend shifts and the extent of these changes.

## Keywords

Artificial Intelligence, Epidemics, Compartment models, Variants, Forecasting, COVID-19

## 1 Introduction

The COVID-19 pandemic underscores the imperative of resilient monitoring systems to effectively navigate global health crises. These systems are indispensable tools for policy-makers, empowering them to manage health emergencies with precision and foresight. Central to their efficacy is the capacity for accurate forecasting, which not only informs strategic decision-making but also enables proactive planning and targeted resource allocation, essential for mitigating the pandemic's impact on public health and societal well-being.

Numerous methodologies exist for predicting the trajectory of epidemics, employing diverse modeling approaches. Machine learning (ML) [39, 28, 4, 35, 31, 17] and deep learning (DL) models [25, 1, 2, 32, 29], including Convolutional Neural Networks (CNNs), Recurrent Neural Networks (RNNs) with Long Short-Term Memory (LSTM) or Gated Recurrent Unit (GRU) cells, and multivariate CNNs have gained prominence. Nonetheless, these data-centric approaches face challenges related to transparency, explainability, and

difficulty in forecasting significant trend changes. These shortcomings are especially problematic when timely and precise forecasting is critical for effective decision-making and intervention.

Conversely, compartmental models are specifically crafted to compute the progression of infections within a population and offer clarity and ease of interpretation for stakeholders like policymakers and healthcare professionals. These models may consider various factors, including vaccinations, variants, different age groups, symptoms, hospitalizations, ICU admissions, undetected infections, and human mobility between regions [8, 24, 14, 13, 5, 26, 27, 23, 38]. Stochastic transmission models, incorporating random variables like individual interactions and variations in infectiousness, were also used to study COVID-19 transmission, providing a nuanced understanding and robust predictions [18, 22]. Despite these advantages, these analytic approaches face several challenges: they depend on assumptions about the system—which may not always be accurate in real-world scenarios—and the parameter estimation is complex, requiring precise data collection for reliable models.

The joint use of data-centric methodologies and analytical approaches, exemplified by the integration of ML techniques with compartmental models, not only augments forecasting accuracy but also bolsters the efficacy of mitigation strategies. This innovative fusion of methodologies, as evidenced in some studies [12, 35, 21, 36], showcases the potential for significantly improving predictive capabilities in epidemic forecasting. By leveraging the strengths of both data-driven and analytical frameworks, researchers can attain a more comprehensive understanding of disease dynamics and thereby enhance the precision of forecasts.

Forecasting epidemic spread aims to predict the percentage of the infected population, fatalities, and hospitalizations at a future point. These metrics stem from complex, nonlinear population dynamics, especially at critical points like peaks or the emergence of new variants. Epidemic dynamics are characterized by widely recognized quantities, such as the basic reproduction number, $R_0$, which expresses the number of secondary infections arising from one single infected individual within a population of susceptible individuals. While $R_0$ shows how fast a disease spreads, its time-dependent counterpart, $R_t$, allows for quantitative evaluation of the infection's course. Such indicators, being specific to the disease,

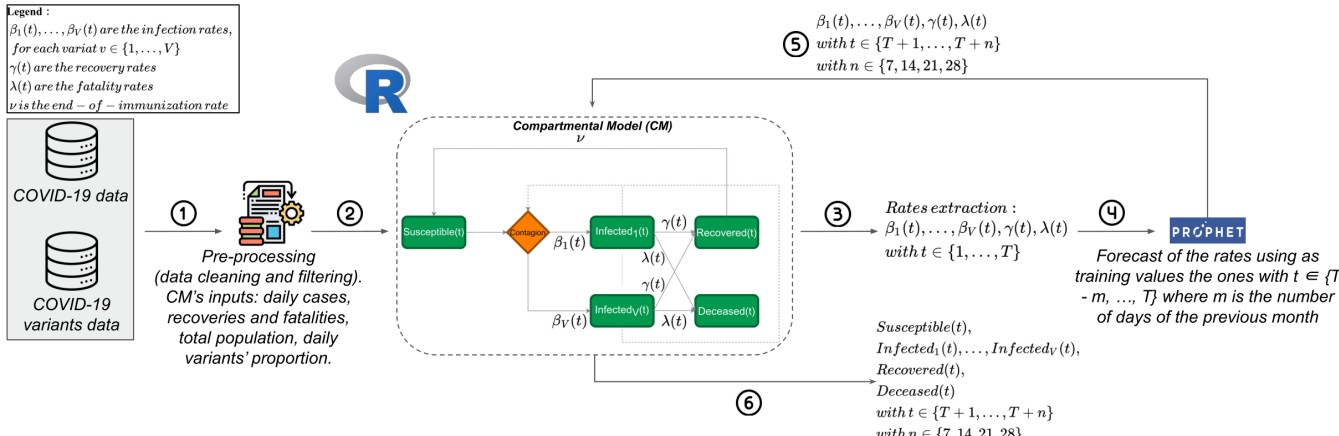

**Figure 1: Schematic representation of Sybil's steps.**

tend to be more stable. Therefore, their future evolution shows a more predictable behavior.

## 1.1 Contributions

We propose Sybil [3], a framework integrating machine learning and compartmental models for better prediction accuracy and explainability. Sybil leverages disease characteristics—like the $R_t$—to project future trends and employs a simple analytical model for infection dynamics. Its strengths include accurate forecasting despite changes in the diffusion process, a reduced need for training data, the ability to study the evolution of multiple variants' infections, reproducible results, and availability as open-source software online.

## 2 Methods

Sybil [3] is an integrated framework designed to deliver accurate and explainable epidemic spread forecasts. It combines a simple compartmental model with a machine learning-based predictive model to forecast infection progression, accounting for multiple virus strains.

Sybil operates in two stages. First, it uses an analytical model to derive critical parameters from surveillance data, specifically the reproductive number over time, $R_t$. Secondly, the data-centric model predicts future parameter values, which are fed back into the analytical model to compute future daily infections—see Figure 1 for a visual overview of Sybil's steps.

Sybil's performance is evaluated by comparing its forecasts against actual surveillance data from Italy, Austria, and U.S. and against predictions obtained from some state-of-the-art approaches, including Prophet [33], ARIMA / SARIMA [6], Neural Prophet [34], LSTM [19] and GRU [10] neural networks, and EpiNow2 [30].

## 2.1 Compartmental analytical model

The analytical component of Sybil is a Susceptible - Infected - Recovered - Deceased - Susceptible (SIRDS) compartmental model described by Equation 1. We reconstruct the evolution of the infection process using surveillance data from the COVID-19 Data

Hub [16, 15]—we used data on cases, recoveries, and fatalities—possibly after a pre-processing phase (step *1)* and *2)* of Figure 1).

$$S(\tilde{t}+1) = S(\tilde{t}) - \boldsymbol{\beta(\tilde{t})}\frac{S(\tilde{t})I(\tilde{t})}{N} + \nu R(\tilde{t})$$
$$I(\tilde{t}+1) = I(\tilde{t}) + \boldsymbol{\beta(\tilde{t})}\frac{S(\tilde{t})I(\tilde{t})}{N} - \boldsymbol{\gamma(\tilde{t})}I(\tilde{t}) - \boldsymbol{\lambda(\tilde{t})}I(\tilde{t}) \quad (1)$$
$$R(\tilde{t}+1) = R(\tilde{t}) + \boldsymbol{\gamma(\tilde{t})}I(\tilde{t}) - \nu R(\tilde{t})$$
$$D(\tilde{t}+1) = D(\tilde{t}) + \boldsymbol{\lambda(\tilde{t})}I(\tilde{t})$$

In this model, the rates are time-dependent—meaning that they may vary at each time step, with the time step corresponding to one day. The only exception is the end-of-immunization rate $\nu$, which is assumed to be constant[1].

Obtaining all the required parameters to solve the equations in Equation 1 is not straightforward as surveillance data does not provide the transition rates—the bold elements of Equation 1. By using Equation 2 (derived from Equation 1), we can estimate the daily infection $\beta(\tilde{t})$, recovery $\gamma(\tilde{t})$, and fatality rates $\lambda(\tilde{t})$, as outlined in step *3)* of Figure 1.

$$\lambda(\tilde{t}) = \frac{D(\tilde{t}+1) - D(\tilde{t})}{I(\tilde{t})}$$
$$\gamma(\tilde{t}) = \frac{R(\tilde{t}+1) - R(\tilde{t}) + \nu R(\tilde{t})}{I(\tilde{t})} \quad (2)$$
$$\beta(\tilde{t}) = \frac{I(\tilde{t}+1) - I(\tilde{t}) + \gamma(\tilde{t})I(\tilde{t}) + \lambda(\tilde{t})I(\tilde{t})}{S(\tilde{t})I(\tilde{t})}N$$

Incorporating variants into the model from Equation 1 requires adding a compartment for each virus strain, creating a $SI^V RDS$ model—for Italy and Austria we used variants' diffusion data from the European Center for Disease Control (ECDC) [11, 20], while for the U.S. we used data from the Centers for Disease Control and Prevention (CDC) [9]. This introduces additional rates: instead of a single infection rate $\beta(\tilde{t})$, there are V different rates $\beta_v(\tilde{t})$, one for each variant at each time step—Equations 3 and 4. Sybil simplifies by assuming that the evolution of each $I_v(\tilde{t})$ compartment is based

---

[1] $\nu = \frac{1}{180}$ since, on average, the immunization due to infection is estimated to be lost after 180 days [37]

on the $I(\tilde{t})$ compartment and the daily proportion of the variant—$I_v(\tilde{t}) = I(\tilde{t})\pi_v(\tilde{t})$, where $\pi_v(\tilde{t})$ is the proportion of infections due to variant $v$ in each time step.

$$I_v(\tilde{t} + 1) = I_v(\tilde{t}) + \boldsymbol{\beta_v(\tilde{t})}\frac{S(\tilde{t})I(\tilde{t})}{N} - \gamma(\tilde{t})I_v(\tilde{t}) - \lambda(\tilde{t})I_v(\tilde{t}) \qquad (3)$$

$$\beta_v(\tilde{t}) = \frac{I_v(\tilde{t} + 1) - I_v(\tilde{t}) + \gamma(\tilde{t})I_v(\tilde{t}) + \lambda(\tilde{t})I_v(\tilde{t})}{S(\tilde{t})I(\tilde{t})}N \qquad (4)$$

## 2.2 Prophet predictive model

The second component of the Sybil framework is Prophet [33], an open-source time series forecasting tool developed by Facebook. It uses an additive model with adjustable parameters, combining statistical modeling and machine learning techniques, including piece-wise linear trends, nonlinear growth, and seasonality adjustments using a Fourier series.

Prophet's flexible approach captures both simple and complex data patterns through three main components: trend ($g(\tilde{t})$), seasonality ($s(\tilde{t})$), and holidays ($h(\tilde{t})$), represented by the equation:

$$y(\tilde{t}) = g(\tilde{t}) + s(\tilde{t}) + h(\tilde{t}) + \epsilon_{\tilde{t}}$$

The error term $\epsilon_{\tilde{t}}$ captures unmodeled changes. Prophet estimates uncertainty in trend forecasts using Markov Chain Monte Carlo (MCMC) to generate many plausible future trajectories. The MCMC samples from the posterior distribution of model parameters, producing a range of possible outcomes used to create multiple forecast trajectories.

From Equations 2 and 4, we extracted the infection rates $\beta_v(\tilde{t})$ for each variant, the recovery rates $\gamma(t)$, and the fatality rates $\lambda(t)$. Using these rates, we applied Prophet to predict the values one, two, three, and four weeks into the future, using the previous month's data for training—step *4)* in Figure 1. We then injected these new values into the SI$^V$RDS model to forecast the evolution of each compartment for the next four weeks—steps *5)* and *6)* in Figure 1.

## 3 Results

Accurate forecasts rely heavily on regular data: linear increases or decreases are easy to predict, while sudden changes are much harder. Outbreaks and peak infection declines often show this unpredictable behavior. The vertical dashed lines in Figure 2 indicate the time point chosen to assess Sybil's accuracy in Italy, Austria and in the state of New York (U.S.). The selected points in Italy and Austria are in the rising phase of an outbreak but near enough to the peak for precise predictions, while the one selected in New York is in the descending phase of the Alpha and the *Other* variant—the latter represents the initial SARS-CoV-2 variant, all the other variants (e.g., Beta, Gamma, Kappa), and some noisy values in the surveillance data. Sybil must accurately capture a change in the function's concavity. Figures from 3 to 7 show the obtained results. For the first scenario in Italy and the scenario in Austria we also compared Sybil's and Prophet's predictions—we chose Prophet since it is a component of Sybil—, and we calculated the Root Mean Squared Error (RMSE) and the standard deviation between the ground truth and the predictions obtained using Sybil and all the considered state-of-the-art approaches—see Table 1.

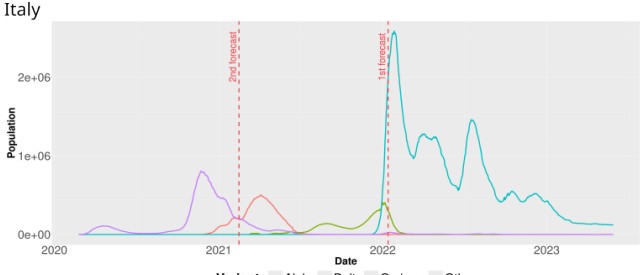

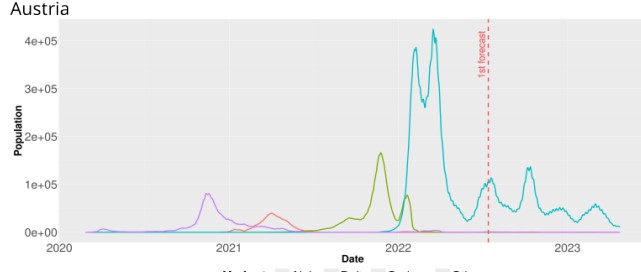

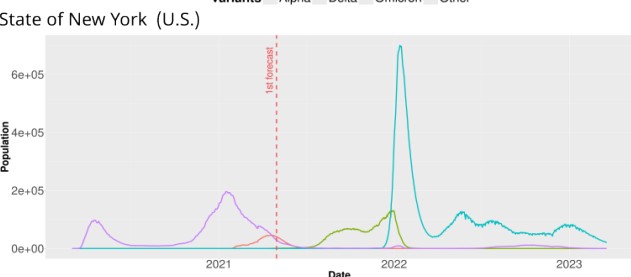

**Figure 2: Daily active cases in Italy, Austria and in the state of New York (U.S.) from February 2020 to May 2023 for the four main SARS-CoV-2 strains.**

In the first scenario in Italy we used data from December 13[th] 2021 to January 13[th] 2022 for training, forecasting daily infections from January 14[th] through February 2022. In Italy, three variants were active during this period: Omicron, Delta, and the *Other* variant. Figure 3-*(a)* shows daily infections for these variants, comparing the ground truth with Sybil's forecasts for one to four weeks. The forecasts are highly accurate for predictions from seven to twenty-one days and slightly anticipate the peak's decline at four weeks. Figure 3-*(b)* contrasts both approaches with the ground truth. Prophet's predictions miss the peak, diverging from real data and failing to provide even a valid qualitative prediction, as they increase while infections decrease.

In the scenario in Austria we consider the period from June 14[th], 2022 to July 14[th], 2022 as training data, and we forecast the daily infections for the period July-August 2022 (starting from July 15[th]). Figure 5 shows that Sybil's predictions demonstrate remarkable accuracy in foreseeing the infection's future trajectory and how well Sybil is able to capture the weekly seasonality presents in the data, which is one of Sybil's strengths.

Finally, the scenario in the state of New York (U.S.) shows that Sybil *i)* can be used for predictions at different levels—country, state (for U.S. states), regional, and city levels—, *ii)* can be used with

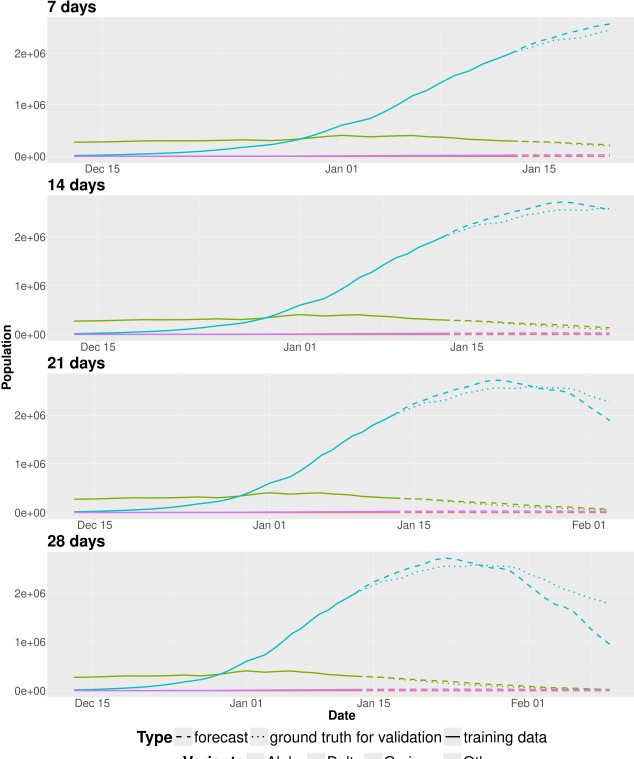

**Figure 3: Evolution of infections using Sybil in the first scenario in Italy (the dashed line shows the prediction, while solid and dotted lines represent the training data and the ground-truth values extracted from the surveillance data, respectively).**

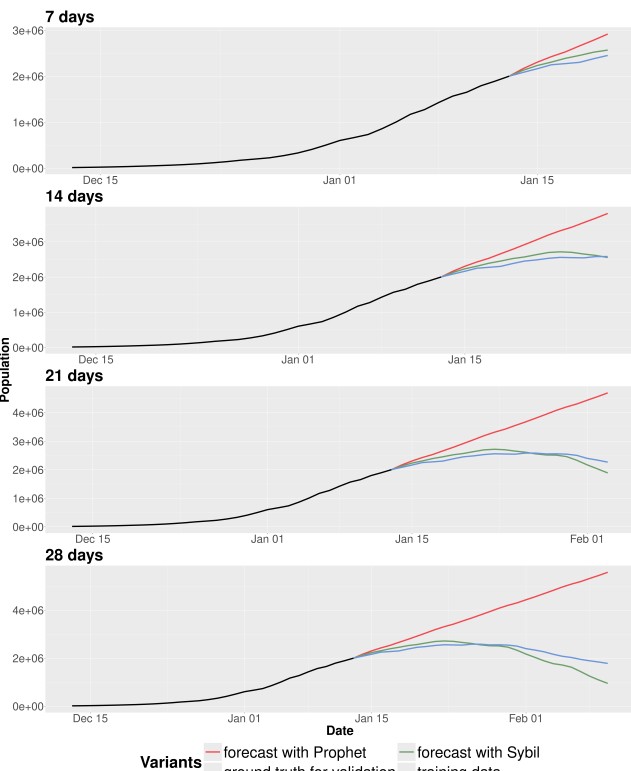

**Figure 4: Comparison between Sybil (green line) and Prophet (red line) in the first scenario in Italy on the number of infections for the Omicron variant using the same period as training data (black line) comparing and contrasting the predictions against the surveillance data for the period spanning the forecasting window (blue line).**

various data sources, *iii)* is able to make predictions on multiple variants, and *iv)* can be used with a fixed recovery rate, to dispense with the often unavailable data on recoveries. Here there are three active variants: Alpha, Delta and the *Other* variant. In this case, we used a fixed recovery rate $\frac{1}{\gamma}$ equal to 14 days [7]—same for each variant. Figure 6 shows that Sybil is also able to capture this flatter trend, especially after one and two weeks.

## 4 Discussion

ML approaches excel at handling complexity by exploiting non-trivial correlations often inaccessible with other tools. However, they require substantial amounts of data, which may not always be available from surveillance. Sybil addresses this issue by using Prophet, a hybrid ML approach combined with simulations, and by not treating virus spread forecasting as a single task. Specifically, by providing the compartmental model with parameters extracted from the real data or forecasts, there is no need to tune the model and estimate the missing parameters, a resource-intensive and situation-specific task, making Sybil easily deployable in new scenarios as long as daily data requirements are met. Additionally, compartmental models provide clear explanations of infection trends, aiding in communication with policy-makers.

In the *Results* section, we presented Sybil's forecasts for Italy, Austria and the state of New York (U.S.), covering periods with a significant changes in daily infection rates. Sybil's predictions

were compared with surveillance data and the plain Prophet application, demonstrating Sybil's superior accuracy. For example, Figure 4 highlights Sybil's precise prediction of a peak that Prophet alone missed. Sybil consistently outperformed other state-of-the-art approaches—see Table 1—, particularly for forecasts spanning two to four weeks. Even with minor or no changes in infection trends, Sybil maintained robust performance.

To set up a continuous monitoring system we have to obtain good predictions also in periods in which there is a new emerging variant or a new exploding outbreak. In particular, in the second scenario in Italy there are three active variants: Alpha, Delta and the *Other* variant. The Alpha variant is ascending while the *Other* variant is descending. Figure 7 shows how the one-week forecast changes moving the training window from February 15[th], 2021 to March 20[th], 2021 by three days and how Sybil is able to capture the future evolution of infections after one week.

Sybil can be applied to make predictions in other countries, as well as at regional and city levels. The methodology is very close to a continuous monitoring system but depends on data availability—e.g., surveillance data available for many countries worldwide reports incorrect data on recoveries or does not report data to devise daily recovery rates. For this reason, we have included the possibility to use a fixed recovery rate, to pre-process missing data, and to work with weekly data.

| Approach | 1 week mean (± std) | 2 weeks mean (± std) | 3 weeks mean (± std) | 4 weeks mean (± std) |
|---|---|---|---|---|
| **Italy** | | | | |
| Sybil | 96883 (± 34583) | **99589** (± 63486) | **190196** (± 186429) | **380341** (± 326962) |
| Prophet | 300266 (± 133638) | 676381 (± 356182) | 1226726 (± 703819) | 1946285 (± 1158953) |
| ARIMA | 196298 (± 106323) | 502508 (± 290262) | 984971 (± 602757) | 1636012 (± 1022713) |
| SARIMA | 196298 (± 106323) | 502508 (± 290262) | 984971 (± 602757) | 1636012 (± 1022713) |
| Neural Prophet | 299890 (± 130724) | 675966 (± 354417) | 1225983 (± 702251) | 1945198 (± 1157424) |
| LSTM | 857350 (± 447682) | - | - | - |
| GRU | 259534 (± 134814) | 768775 (± 470723) | 2362125 (± 1714411) | - |
| EpiNow2 | **64997** (± 63032) | 2417232 (± 161043) | 2429904 (± 147212) | 2322897 (± 245505) |

| Approach | 1 week mean (± std) | 2 weeks mean (± std) | 3 weeks mean (± std) | 4 weeks mean (± std) |
|---|---|---|---|---|
| **Austria** | | | | |
| Sybil | 4383 (± 2052) | **5237** (± 4589) | **10463** (± 9874) | **12605** (± 10515) |
| Prophet | 3830 (± 969) | 7230 (± 7104) | 21879 (± 17776) | 33265 (± 23837) |
| ARIMA | 8689 (± 2818) | 23719 (± 13307) | 46740 (± 28039) | 66184 (± 38009) |
| SARIMA | 8689 (± 2818) | 23719 (± 13307) | 46740 (± 28039) | 66184 (± 38009) |
| Neural Prophet | 4027 (± 1149) | 7145 (± 7055) | 21636 (± 17674) | 32931 (± 23698) |
| LSTM | 3245 (± 2870) | 7702 (± 6778) | 22690 (± 17181) | 33687 (± 22817) |
| GRU | 4434 (± 3173) | 5931 (± 5847) | 19115 (± 15538) | 28769 (± 20578) |
| EpiNow2 | **2696** (± 2695) | 105529 (± 5070) | 97197 (± 14127) | 89894 (± 18865) |

**Table 1: RMSE with std between the ground truth used for validation and the obtained forecast with Sybil and the plain use of different state-of-the-art approaches for the first scenario in Italy and the scenario in Austria. Values in bold represent minimum values, while underlined values represent values close to minimum values. For LSTM and GRU we do not report the errors in some cases because they are too high.**

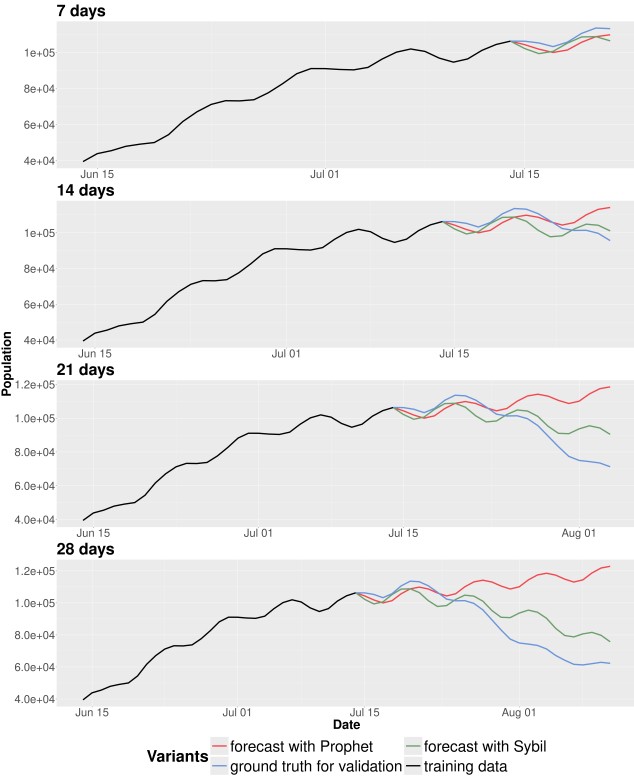

**Figure 5: Comparison between Sybil (green line) and Prophet (red line) in the scenario in Austria on the number of infections for the Omicron variant using the same period as training data (black line) comparing and contrasting the predictions against the surveillance data for the period spanning the forecasting window (blue line).**

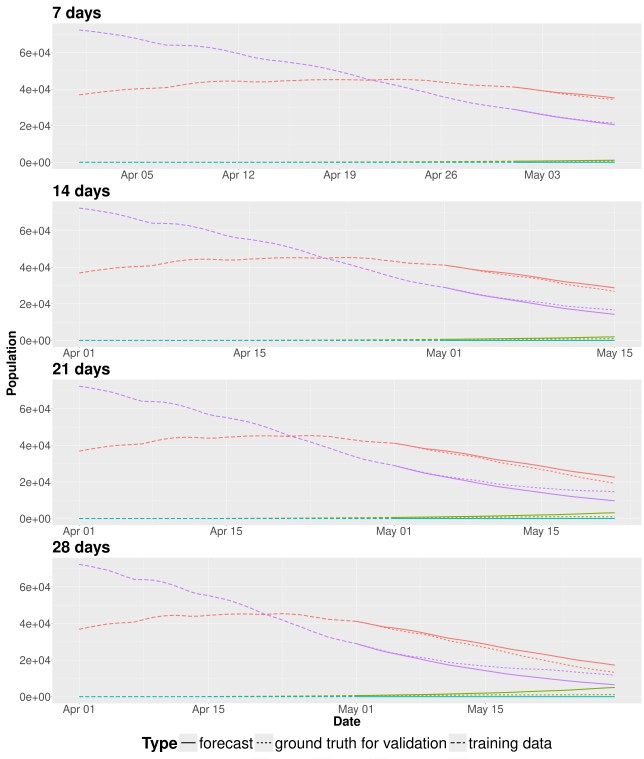

**Figure 6: Evolution of infections using Sybil in the scenario in the state of New York (the dashed line shows the prediction, while solid and dotted lines represent the training data and the ground-truth values extracted from the surveillance data, respectively).**

## 5 Conclusion

The COVID-19 pandemic underscores the critical need for advanced tools to monitor and forecast infections. This paper presents Sybil, a cutting-edge framework seamlessly integrating machine learning and compartmental models. Sybil provides reliable, replicable, and explainable forecasts, validated through extensive experimentation. Sybil accurately predicts peaks and emerging outbreaks and integrates variants, aiding policy-makers. By using only data from the previous month, Sybil reduces the need for extensive training data, enhancing computational efficiency. By combining data-centric and analytic approaches, Sybil overcomes inherent limitations, making it a versatile tool not only for COVID-19 but also for other diseases, empowering policy-makers to respond swiftly to emerging threats.

## 6 Future works

Possible future directions include trying different ML component instead of Prophet—such as Neural Prophet [34], LSTM [19], GRU [10], or other alternatives—, including vaccinations and hospitalizations in the compartmental model, using stochastic simulations instead of deterministic ones, and improving explainability.

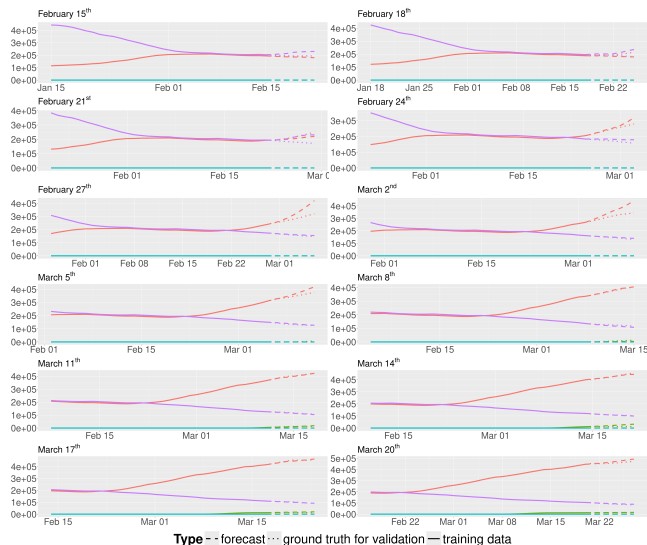

**Figure 7: Evolution of $I_v(\tilde{t})$ compartments using Sybil in the second scenario in Italy starting the forecast from February 15th, 2021 and moving the training window by three days (the dashed line shows the prediction, while solid and dotted lines represent the training data and the ground-truth values extracted from the surveillance data, respectively). All plots refer to a forecast one week into the future.**

## Acknowledgments

D.B. is a Ph.D. student enrolled in the National Ph.D. in Artificial Intelligence, XXXVII cycle, health and life sciences course organized by Università Campus Bio-Medico di Roma.

## Availability of materials and data

COVID-19 data used in this study are available in the COVID19 R library [16, 15] and in the ECDC and CDC variants data [11, 20, 9]. The code is available at https://github.com/daniele-baccega/sybil-forecasting.

## Funding

This work was supported by grants from "Ripresa delle attività socio-economiche e delle scuole: modelli per la progettazione e supporto di linee guida per la convivenza con il Covid-19" (Cod. ROL 73459, 2020, PI Matteo Sereno), project funded by CRT foundation, and from TED2021-131264B-I00 (SocialProbing) funded by MCIN/AEI/10.13039/501100011033 and the European Union "NextGenerationEU"/PRTR.

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
