# OpenReview forum: "Enhancing COVID-19 Forecasting Precision through the Integration of Compartmental Models, Machine Learning and Variants"
_KDD.org/2024/Workshop/epiDAMIK — KDD 2024 Workshop epiDAMIK_

### Official Review · Reviewer_hRdd · 2024-06-21
**Fundamental limitation for practical use**

**Rating:** 2
**Confidence:** 3

**Review:**

Positives:
- This paper contributes to a body of literature that addresses a need in the community to focus on methods that combine trend extension (ML) methods with analytical methods that reflect how data and respiratory illness disease progression have changed (in addition to other works both cited and uncited here)
- This paper explicitly highlights the challenges in modeling multiple variants
- This paper is structured and proofread

Fundamental Concerns:
- This retrospective analysis likely uses revised data vs. as-of data (especially regarding the variants). Because COVID data was revised, these results, which, as far as I can tell in this paper, do not reflect real-time data delays. Thus their practical application may be limited and does not support claims like , "Sybil provides reliable, replicable, and explainable forecasts, validated through extensive experimentation. Sybil accurately predicts peaks and emerging outbreaks and integrates variants, aiding policymakers."
- In addition, the evaluation does not support all of these claims with the definitiveness they are stated- there were many more waves during the pandemic, and additional validation is also important across those peaks. Further the direct comparison with the baseline methods is not as interesting as the community knows that simple trend extension did not capture waves during the pandemic. For example, comparing to a baseline model that was given more data across time to see if it can learn the relationships that the SEIR model captures implicitly would reveal something novel about the relative benefits of Sybill.
- "These disease-specific indicators are generally stable and more predictable." I think there is growing contention about how quickly R_t changes due to the noisiness in deriving this value from surveillance data
- Multiple epi-modeling community discussions center around the noisiness and non-representativeness of using modeling parameters for forecasting [the most recent reference is the May CSTE Forecasting Workgroup Call], so the claim, "This approach eliminates the need for parameter tuning," may not be accurate in practice.

 Stylistic Suggestion:
- Some discussion about Italian public health surveillance data could contextualize if this approach would work in other regions/settings

---

### Official Review · Reviewer_kF9h · 2024-06-28
**Additional experimental evaluation could strengthen the paper.**

**Rating:** 3
**Confidence:** 4

**Review:**

Summary:
- The authors use Prophet, a machine learning-based time series forecasting model, to estimate future transmission parameters that are used within a SIRD compartmental framework to predict COVID-19 within a four-week period.

Strengths:
- Combines the forecasting capability of machine learning-based method and the explainability of more traditional compartmental methods.
- Demonstrates better forecasting performance of their Sybil model for the longer four-week prediction period in comparison with appropriate baselines such as Neural Prophet and EpiNow2.

Major Weaknesses:
- Limited evaluation:
    - The authors only evaluated performance at one time point, near the first peak of the Omicron variant. Since the author’s motivation is to test the model’s ability to predict sudden changes, analysis of additional time points at later peaks of the Omicron variant could strengthen the results. If the authors only evaluate this time point, more explanation is needed as to why other time points are not chosen.
    - In addition, the current prediction period is too short to evaluate the long-term forecasting capabilities of the model fully. The concavity of the peak is forecasted, yet the rate of decline cannot be fully captured within four weeks. Prior works have evaluated up to 90 days or longer for COVID-19 forecasting.
    - Lastly, one of the novelty components of the model is its ability to capture multiple variants. The author’s current evaluation occurs at a time in which the Delta variant is negligible in comparison with the Omicron variant. Perhaps evaluation around 2021 with the decline of the initial SARS-CoV-2 variant and the increase of the Alpha and Delta variants could be more compelling.
- Limited external validation: Alternatively, the authors can try to benchmark the model on other datasets. The work currently only utilizes data from Italy. External validation of Sybil vs. Prophet on data from another country can strengthen the experimental findings of the paper.

Additional Comments:
- Figure 1 is a bit hard to read. In Step 1, it’s unclear what data preprocessing entails, and what type of features are used as input to the compartmental model in Step 2. Step 3 and Step 5 need text descriptions (e.g. infection, recovery, fatality rates), since the symbols are not defined until later in the Methods section. Font size could be increased. The actual description of the model in Section 2 is easy to follow, it’s just the figure itself that’s difficult to understand.
- Perhaps use scientific notation for Table 1? There is no need for the full precision of the numbers.
- In Figure 3b, we see that Prophet fails to capture the downward trend in comparison with Sybil. Although the authors do not necessarily have to plot the other baselines, more discussion of the forecast deviation of the other baselines would be welcome in the results section. For instance, the LSTM and GRU errors are omitted “because they are too high”. What is the reason for this? Are the forecast trajectories not declining after the peak, or are the models failing to train completely on the given data?
- For future work, the authors could look into incorporating counterfactual predictions based on different policies (e.g. vaccinations, social distancing) into the framework.

---

### Official Review · Reviewer_4iC6 · 2024-06-30
**Bolstering forecasts through the inclusion of compartmental models and variants**

**Rating:** 4
**Confidence:** 3

**Review:**

### Summary
This paper contributes to strengthening disease incidence forecasting models through adding an additional structural element - incorporating information from compartmental models. In addition, they consider the presence of multiple variants, which helps the capability of the forecast. They evaluate this method through Italian data measuring cases, deaths, and variants. They mainly focus on the performance of such forecasts at times where the course of the outbreak changes significantly.


### Strong Points
- The paper makes thorough reference to the sources of data they use.
- The inclusion of compartmental models in the framework adds better explainability - a great follow-up to this work may be a paper which focuses solely on the interpretation of forecasts and how they may be explained.


### Weak Points
- In section 2.1, it is stated "the rate are time-dependent..." The paper does not discuss the potential of overfitting given this paradigm and the presence of (effectively) a large number of parameters.
- The range of settings presented as experiments are somewhat limited. More locations and periods would be helpful.

### Suggestions
- Include a subsection of the introduction making the contributions of this work more explicit
- In the introduction, it is stated that "epidemic dynamics is often measured by ... R0". This seems like an oversimplification, especially for a forecasting problem. Perhaps remove this sentence or replace it with a more appropriate description of R0.
- Additional speculation on why Sybil predicted the peak Prophet alone missed would be nice. Is this capability a direct result of the inclusion of the compartmental model? What if only one variant was included in Sybil?

---

### Official Review · Reviewer_X2Ge · 2024-07-01

**Rating:** 4
**Confidence:** 5

**Review:**

### Summary

This work proposes a method, namely Sybil, that integrates compartmental models with neural network-based prediction models for predicting infection cases during epidemic evolution. Specifically, Sybil first uses an analytical model to derive the parameters from historical data, such as the reproductive number over time $R_t$. Then, Sybil fits a machine learning model to predict future parameter values. The values are fed back into the analytical model to compute future daily infections. This work evaluates the proposed method on COVID-19 infection cases in Italy. The results show that the proposed method achieves better estimation error than existing methods (including LSTM, GRU, ARIMA and SARIMA).

### Strengths
- This work proposes a method that integrates compartmental models with neural network-based prediction models.
- This work provides a case study on the COVID-19 spreading in Italy and shows the benefits against several existing methods.

### Weakness.
- It would be better to give insights into which case, the proposed method works better for the existing method, such as Prophet.
- The Prophet predictive model can be further described with more details.
- It would be great if the data for the case study can be open-sourced for future studies.